# CORRECTIVE RETRIEVAL AUGMENTED GENERATION

## ABSTRACT

Large language models (LLMs) inevitably exhibit hallucinations since the accuracy of generated texts cannot be secured solely by the parametric knowledge they encapsulate. Although retrieval-augmented generation (RAG) is a practicable complement to LLMs, it relies heavily on the relevance of retrieved documents, raising concerns about how the model behaves if retrieval goes wrong. To this end, we propose the **C**orrective **R**etrieval **A**ugmented **G**eneration (CRAG) to improve the robustness of generation. Specifically, a lightweight retrieval evaluator is designed to assess the overall quality of retrieved documents for a query, returning a confidence degree based on which different knowledge retrieval actions can be triggered. Since retrieval from static and limited corpora can only return suboptimal documents, large-scale web searches are utilized as an extension for augmenting the retrieval results. Besides, a decompose-then-recompose algorithm is designed for retrieved documents to selectively focus on key information and filter out irrelevant information in them. CRAG is plug-and-play and can be seamlessly coupled with various RAG-based approaches. Experiments on four datasets covering short- and long-form generation tasks show that CRAG can significantly improve the performance of RAG-based approaches.

## 1 INTRODUCTION

Large language models (LLMs) have attracted increasing attention and exhibited impressive abilities to understand instructions and generate fluent language texts (Brown et al., 2020; Ouyang et al., 2022; Touvron et al., 2023a). Nevertheless, LLMs inevitably manifest hallucinations (Ji et al., 2023) due to their struggle with factual errors (Mallen et al., 2023; Min et al., 2023) and inability to secure the accuracy of generated texts solely by the parametric knowledge they encapsulate (Zhang et al., 2023b; Muhlgay et al., 2023).

Prior research has introduced the retrieval techniques to incorporate the knowledge relevant to input and augment generation, as exemplified by retrieval-augmented generation (RAG) (Lewis et al., 2020). In this framework, the input to models is augmented by prepending relevant documents that are retrieved from an external knowledge corpus (Guu et al., 2020). While RAG serves as a practicable complement to LLMs, its effectiveness is contingent upon the relevance and accuracy of the retrieved documents (Li et al., 2022; Tan et al., 2022). The heavy reliance of generation on the retrieved knowledge raises significant concerns about the model's behavior and performance in scenarios where retrieval may fail or return inaccurate results (Shi et al., 2023). As Figure 1 shows that a low-quality retriever is prone to introducing a substantial amount of irrelevant information, impeding the models from acquiring accurate knowledge and potentially misleading them, resulting in issues such as hallucinations (Zhang et al., 2023b). However, most conventional RAG approaches indiscriminately incorporate the retrieved documents, regardless of whether these documents are relevant or not (Rony et al., 2022). Furthermore, current methods mostly treat complete documents as reference knowledge both during retrieval and utilization. But a considerable portion of the text within these retrieved documents is often non-essential for generation, which should not have been equally referred to and involved in RAG.

On account of the above issues, this paper particularly studies the scenarios where the retriever returns inaccurate results. A method named **C**orrective **R**etrieval-**A**ugmented **G**eneration (CRAG) is proposed to self-correct the results of retriever and improve the utilization of documents for augmenting generation. A lightweight retrieval evaluator is designed to assess the overall quality of retrieved documents for a query. This serves as a crucial component in RAG, contributing to

informative generation by reviewing and evaluating the relevance and reliability of the retrieved documents. A confidence degree is quantified based on which different knowledge retrieval actions of {Correct, Incorrect, Ambiguous} can be triggered. For the latter two actions, large-scale web searches (Piktus et al., 2021; Komeili et al., 2022) are integrated as a strategic extension, since retrieval from static and limited corpora can only return sub-optimal documents in terms of scope and diversity. This augmentation is implemented to broaden the spectrum of retrieved information, harnessing the expansive and dynamic nature of the web to complement and enrich the initially obtained documents. Furthermore, to eliminate redundant contexts contained in retrieved documents that are unhelpful for RAG, a decompose-then-recompose algorithm is meticulously crafted throughout the retrieval and utilization process. This algorithm ensures the refinement of retrieved information, optimizing the extraction of key insights and minimizing the inclusion of non-essential elements, thereby enhancing the utilization of retrieved data.

CRAG is plug-and-play and experimentally implemented into RAG (Lewis et al., 2020) and Self-RAG (Asai et al., 2024) for demonstrating its adaptability to RAG-based approaches. Results on four datasets of PopQA (Mallen et al., 2023), Biography (Min et al., 2023), Pub Health (Zhang et al., 2023a), and Arc-Challenge (Bhakthavatsalam et al., 2021) show that CRAG can significantly improve the performance of standard RAG and state-of-the-art Self-RAG, demonstrating its generalizability across both short- and long-form generation tasks. To facilitate others to reproduce our results, we will publish all source code later.

In summary, our contributions in this paper are three-fold: 1) This paper studies the scenarios where the retriever returns inaccurate results and, to the best of our knowledge, makes the first attempt to design corrective strategies for RAG to improve its robustness. 2) A plug-and-play method named CRAG is proposed to improve the ability of automatic self-correction and efficient utilization of retrieved documents. 3) Experimental results extensively demonstrate CRAG's adaptability to RAG-based approaches and its generalizability across short- and long-form generation tasks.

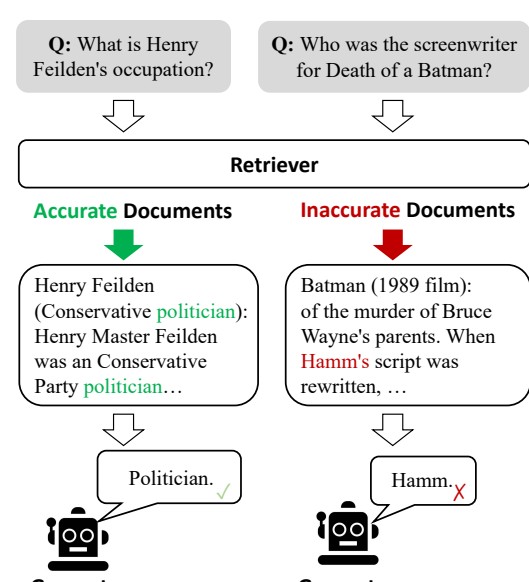

Figure 1: The examples show that a low-quality retriever is prone to introducing a substantial amount of irrelevant information, impeding the generators from acquiring accurate knowledge and potentially misleading them.

## 2 RELATED WORK

**Hallucinations of LLMs** Although LLMs have exhibited impressive abilities to understand instructions and generate fluent language texts (Bang et al., 2023; Qin et al., 2023; Zhong et al., 2023), one of the most severe issues that LLMs have still been struggling with is hallucinations. As many studies found (?Shuster et al., 2021), either outdated information or incorrect knowledge that is activated would seriously result in hallucinations. Large-scale unregulated training data collection, low proportion of high-quality sampling data, imperfection of data allocation in the input space, and many other realistic factors could impact the LLMs and exacerbate the problems. Thus, it is obvious that the lack of accurate and specific knowledge can lead to misleading or even inaccurate generation, which will severely hurt the experience of users in most practical applications.

**Retrieval-Augmented Generation** RAG (Lewis et al., 2020; Guu et al., 2020) is regarded as a useful method to address the issues above, which enhances the input questions of generative LMs with retrieved documents. It usually provides an extra knowledge source from a specific corpus, i.e., Wikipedia, which greatly improves the performance of LMs in a variety of tasks, especially in the knowledge-intensive ones. The proposed methods generally leverage information retrieval

to supply documents containing relevant knowledge for generative LLMs. Earlier studies adopt either sparse or dense retrievers at the front end of a pre-trained language model that specializes in response generation. Despite this, the methods above usually ignore a question, *what if the retrieval goes wrong?* Since the purpose of introducing a retrieval is to secure that generative LMs can obtain relevant and accurate knowledge. If retrieved documents are irrelevant, the retrieval system can even exacerbate the factual error that LMs make.

**Advanced RAG**   Many advanced approaches have been developed from the original RAG in recent years  (Zhang et al., 2024; Kim et al., 2024; Wang et al., 2024; Liu et al., 2024). Considering that retrieval is sometimes unnecessary for some queries, conversely, responses without retrieval are even more accurate in many situations. Self-RAG (Asai et al., 2024) is proposed to selectively retrieve knowledge and introduce a critic model to decide whether to retrieve. Yoran et al. (2024) designed an NLI model to identify the irrelevant context and improve robustness. SAIL (Luo et al., 2023) is tuned on instructions to insert retrieved documents before instructions. While Toolformer (Schick et al., 2023) is pre-trained for calling APIs such as Wikipedia. In addition, in some long-text generation tasks, external knowledge is needed more than once, and when to retrieve should be concerned. Jiang et al. (2023) actively anticipate future content and decide when and what to retrieve in long-form generation.

Compared with recent studies (Schick et al., 2023; Luo et al., 2023; Asai et al., 2024) that are the most relevant to our work, a main difference should be highlighted. These approaches target on exploiting retrieval as a useful tool to augment generation or whether retrieval is necessary, while this study particularly studies the scenarios where the retriever returns inaccurate results. To the best of our knowledge, this paper makes the first attempt to explore and design corrective strategies for RAG to improve its robustness of generation.

## 3 TASK FORMULATION

Following previous work (Lewis et al., 2020; Asai et al., 2024), given input $\mathcal{X}$ and an accessible corpus containing a large amount of knowledge documents $\mathcal{C} = \{d_1, ..., d_N\}$, the system is expected to generate the output $\mathcal{Y}$. The entire framework is usually divided into a retriever $\mathcal{R}$ and a generator $\mathcal{G}$. The retriever $\mathcal{R}$ aims to retrieve the top-$\mathcal{K}$ documents $\mathcal{D} = \{d_{r_1}, ..., d_{r_k}\}$ that are relevant to the input $\mathcal{X}$ from the corpus $\mathcal{C}$. Based on the input $\mathcal{X}$ and the retrieved results $\mathcal{D}$, the generator $\mathcal{G}$ is responsible for generating the output $\mathcal{Y}$. This framework can be formulated as:

$$P(\mathcal{Y}|\mathcal{X}) = P(\mathcal{D}|\mathcal{X})P(\mathcal{Y}, \mathcal{D}|\mathcal{X}). \tag{1}$$

It shows that the retriever and generator are seamlessly coupled, exhibiting low risk tolerance. Any unsuccessful retrieval can result in an unsatisfactory response, regardless of the impressive abilities of the generator. This is exactly the focus of this paper to improve the robustness of generation.

## 4 CRAG: CORRECTIVE RETRIEVAL AUGMENTED GENERATION

### 4.1 OVERVIEW OF MODEL INFERENCE

Figure 2 and Algorithm 1 present an overview of CRAG at inference, which designs corrective strategies to improve the robustness of generation. Given an input query and the retrieved documents from any retriever, a lightweight retrieval evaluator is constructed to estimate the relevance score of retrieved documents to the input query (Section 4.2). The relevance score is quantified into a total of three confidence degrees and then triggered the corresponding actions: {Correct, Incorrect, Ambiguous} (Section 4.3). If the action Correct is triggered, the retrieved documents will be refined into more precise knowledge strips. This refinement operation involves knowledge decomposition, filter, and recomposition (Section 4.4). If the action Incorrect is triggered, the retrieved documents will be discarded. Instead, web searches are resorted to and regarded as complementary knowledge sources for corrections (Section 4.5). Eventually, when it cannot confidently make a correct or incorrect judgment, a soft and balanced action Ambiguous which combines both of them is triggered. After optimizing the retrieval results, an arbitrary generative model can be adopted.

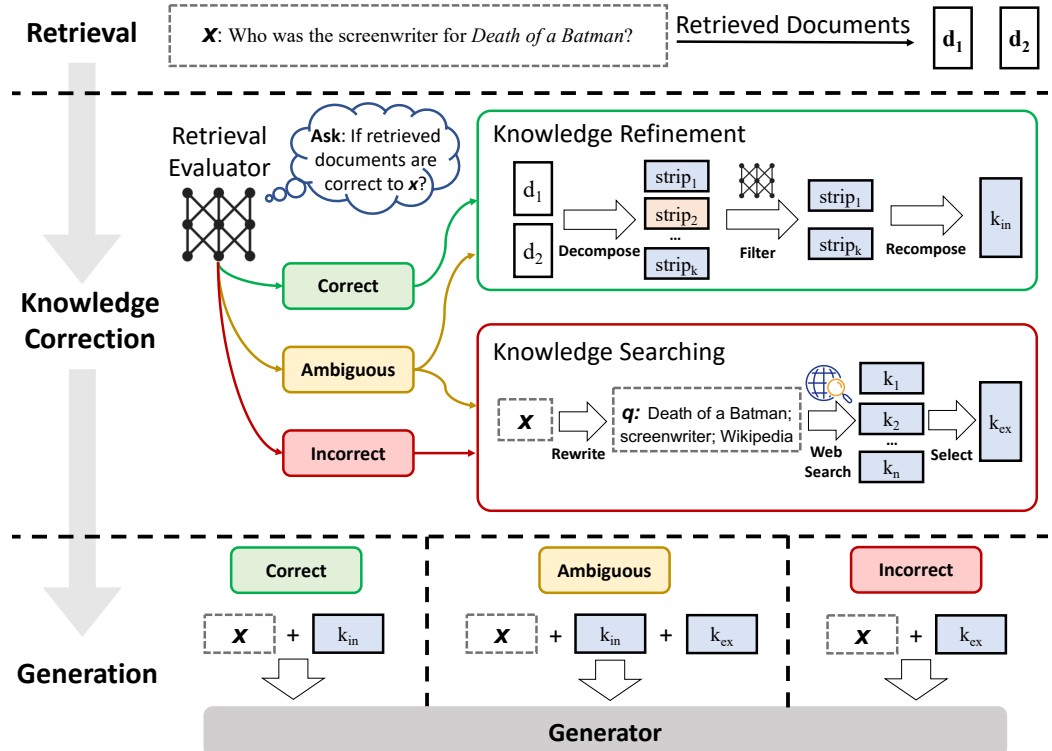

Figure 2: An overview of the proposed CRAG at inference. A retrieval evaluator is constructed to evaluate the relevance of the retrieved documents to the input, and estimate a confidence degree based on which different knowledge retrieval actions of {Correct, Incorrect, Ambiguous} can be triggered.

## 4.2 RETRIEVAL EVALUATOR

It is natural to wonder whether the retrieved documents are accurate or not before using them, which is significant since irrelevant or misleading messages can be identified in this way. The accuracy of the retrieval evaluator undeniably plays a pivotal role in shaping the overall system performance, as it influences the outcomes of subsequent processes. Our objective is to correct the retrieved documents if they are irrelevant. Specifically, T5-large (Raffel et al., 2020) is adopted for initializing the retrieval evaluator and fine-tuned. Its parameter size is much smaller than the most current LLMs (Touvron et al., 2023a;b; Chowdhery et al., 2023; Anil et al., 2023; Brown et al., 2020; Ouyang et al., 2022; OpenAI, 2023). To ensure all experimental results were comparable with Self-RAG (Asai et al., 2024), the same retrieval results through Contriever (Izacard et al., 2022) provided by Self-RAG were also adopted in our experiments. The relevance signals for fine-tuning the evaluator can be collected from the existing datasets. For example, PopQA (Mallen et al., 2023) provides the golden subject wiki title from wikipedia for each question. We can use that to track a not 100% relevant but rather high-quality passage. We utilized that as the relevance signals for fine-tuning the retrieval evaluator.[1] On the other hand, the negative samples for fine-tuning were all randomly sampled from the retrieval results, which are rather similar to the input query but not relevant. More details about this fine-tuning step can be referred to in Appendix B.3. For every question, there are generally 10 documents retrieved. The question is concatenated with each single document as the input, and the evaluator predicts the relevance score for each question-document pair individually. We also tried to prompt ChatGPT to identify the retrieval relevance for comparison, but it underperforms as elaborated in Section 5.5. Based on these calculated relevance scores, a final judgment is made as to whether the retrieval is correct or not associated with the action trigger. In our proposed framework, the retrieval quality is evaluated at a relatively low cost without the need to have access to large and expensive LLMs. Compared with the critic model of

---

[1]https://huggingface.co/datasets/akariasai/PopQA

---

**Algorithm 1:** CRAG Inference

---

**Require:** $E$ (Retrieval Evaluator), $W$ (Query Rewriter), $G$ (Generator)
**Input** : $x$ (Input question), $D = \{d_1, d_2, ..., d_k\}$ (Retrieved documents)
**Output :** $y$ (Generated response)
1   $score_i = E$ evaluates the relevance of each pair $(x, d_i)$, $d_i \in D$
2   **Confidence** = Calculate and give a final judgment based on $\{score_1, score_2, ...score_k\}$
    // **Confidence** has 3 optional values: [CORRECT], [INCORRECT] or [AMBIGUOUS]
3   **if** *Confidence == [CORRECT]* **then**
4      Internal_Knowledge = Knowledge_Refine($x$, $D$)
5      $k$ = Internal_Knowledge
6   **else if** *Confidence == [INCORRECT]* **then**
7      External_Knowledge = Web_Search($W$ Rewrites $x$ for searching)
8      $k$ = External_Knowledge
9   **else if** *Confidence == [AMBIGUOUS]* **then**
10      Internal_Knowledge = Knowledge_Refine($x$, $D$)
11      External_Knowledge = Web_Search($W$ Rewrites $x$ for searching)
12      $k$ = Internal_Knowledge + External_Knowledge
13   **end**
14   $G$ predicts $y$ given $x$ and $k$

---

Self-RAG (Asai et al., 2024) that instruction-tuned LLaMA-2 (7B), the evaluator designed in CRAG demonstrates the advantages of being quite lightweight (0.77B).

### 4.3 ACTION TRIGGER

To correct the irrelevant documents and refine the target documents as needed, actions should be executed discriminately. Based on the aforementioned confidence score for each retrieved document, three types of actions are designed and triggered accordingly where the upper and lower thresholds are set. If the confidence score is higher than the upper threshold, the retrieved document is identified as Correct, while identified as Incorrect if below the lower threshold. Otherwise, a more soft and intermediate action, i.e., Ambiguous is executed. Each retrieved document is conducted individually and integrated eventually.

**Correct**   Here, a retrieval is assumed Correct when the confidence score of *at least one retrieved document* is higher than the upper threshold. If so, it means that there are relevant documents in the retrieved results, and the knowledge from the retrieval results is supposed to be more reliable and accurate. However, even if a relevant document can be found, there is inevitably some noisy knowledge strips in this document. To extract the most critical knowledge strips within this document, a knowledge refinement method is further designed which will be elaborated in Section 4.4.

**Incorrect**   Besides, a retrieval is assumed Incorrect when the confidence scores of *all retrieved documents* are below the lower threshold. This indicates that all retrieved documents are considered irrelevant, which are unhelpful for generation. Once the knowledge from the retrieval results is judged to be inaccurate, it is unwise to still get stuck in it, which is likely to result in fabricated facts. Therefore, we need to seek new sources of knowledge for correction. Here, web search is introduced to search from the Internet as elaborated in Section 4.5. This corrective action helps overcome the embarrassing challenge where no reliable knowledge can be referred to.

**Ambiguous**   Except for the above two situations, the remaining will be assigned to an intermediate action of Ambiguous. This generally occurs when the accuracy of the retrieval is hard to distinguish and the evaluator gives an intermediate score. Since the retrieval evaluator is not confident in its judgment, both types of processed knowledge in Correct and Incorrect are combined to complement each other. Implementing such a moderating and soft strategy can significantly contribute to strengthening the robustness and resilience of the system, fostering a more adaptable framework for optimal performance.

**Discussion** Preliminary experiments of employing only the `Correct` and `Incorrect` actions show that the efficacy of CRAG was easily affected by the accuracy of the retrieval evaluator. The reason might be the distinct knowledge switch for all input cases, regardless of the level of confidence in their judgment. The design of the `Ambiguous` action significantly helps to mitigate the dependence on the accuracy of the retrieval evaluator.

### 4.4 KNOWLEDGE REFINEMENT

Given a retrieved relevant document, a decompose-then-recompose knowledge refinement method is designed to further extract the most critical knowledge strips in it. To obtain fine-grained retrieval results, we segmented the retrieved results into internal strips. If a retrieved result is as short as one or two sentences, it is regarded as an individual strip, otherwise, retrieval documents are required to be split into smaller units which generally consist of a few sentences according to the total length. The scale is assumed to include an independent piece of information, and the filtering is based on the segments. Then, the retrieval evaluator fine-tuned in Section 4.2 is employed to calculate the relevance score of each knowledge strip. Based on these scores, irrelevant knowledge strips are filtered out, while relevant ones are recomposed via concatenation in order, namely internal knowledge.

### 4.5 WEB SEARCH

It would be more intelligent if a system itself could determine that its existing knowledge corpus could not solve the problem well and turn to additional external knowledge for help. On the contrary, even if a system knows that the existing knowledge cannot solve the problem, but still sticks to the limited knowledge corpus, it will only give a fabricated fact in the end, which is called hallucination. Therefore, it is extremely important to seek complementary external knowledge if the retrieved results are all assumed irrelevant, and we consider a system that knows what it doesn't know and what it cannot answer to be more intelligent than one that clings to limited knowledge and is incapable of seeking external knowledge. Since retrieval from static and limited corpora can only return sub-optimal documents in terms of scope and diversity, large-scale web searches (Piktus et al., 2021; Komeili et al., 2022) are integrated as a strategic extension of RAG. Specifically, the inputs are rewritten into queries composed of keywords by ChatGPT to mimic the daily usage of search engine. The prompt for rewriting is shown in Appendix A. In CRAG, a public and accessible commercial web search API is adopted to generate a series of URL links for every query. Considering that knowledge from large-scale web searches could introduce biases or unreliable information, authoritative and regulated web pages like Wikipedia are preferred, which can significantly help mitigate these issues. Moreover, we utilize the URL links to navigate web pages, transcribe their content, and employ the same knowledge refinement method as Section 4.4 to derive the relevant web knowledge, namely external knowledge.

## 5 EXPERIMENTS

We conducted experiments to extensively demonstrate CRAG's adaptability to RAG-based approaches and its generalizability across both short- and long-form generation tasks.

### 5.1 TASKS, DATASETS AND METRICS

CRAG was evaluated on four datasets, including **PopQA** (Mallen et al., 2023) (*short*-form generation), **Biography** (Min et al., 2023) (*long*-form generation), **PubHealth** (Zhang et al., 2023a) (*true-or-false* question), and **Arc-Challenge** (Bhakthavatsalam et al., 2021) (*multiple-choice* question). Following previous work, accuracy was adopted as the evaluation metric for PopQA, PubHealth, and Arc-Challenge. FactScore (Min et al., 2023) was adopted as the evaluation metric for Biography. Readers can refer to Appendix B.1 for more details. The same metrics are used because our proposed method is comparable to previous studies, since we used the same retrieval results as previous work. The difference lies in that our motivation is to improve the retrieval quality by correcting the retrieval results that the system judges to be of low quality. This can be analogous to RAG's augmentation to standalone parameterized language models and we further augment RAG with corrective strategies.

Table 1: Overall evaluation results on the test sets of four datasets. Results are separated based on the generation LLMs. **Bold** numbers indicate the best performance among all methods and LLMs. **Gray-colored** bold scores indicate the best performance using a specific LLM. * indicates the results reproduced by us, otherwise results except ours are cited from their original papers.

| Method | PopQA (Accuracy) | Bio (FactScore) | Pub (Accuracy) | ARC (Accuracy) |
|---|---|---|---|---|
| *LMs trained with propriety data* | | | | |
| LLaMA2-c$_{13B}$ | 20.0 | 55.9 | 49.4 | 38.4 |
| Ret-LLaMA2-c$_{13B}$ | 51.8 | 79.9 | 52.1 | 37.9 |
| ChatGPT | 29.3 | 71.8 | 70.1 | **75.3** |
| Ret-ChatGPT | 50.8 | - | 54.7 | **75.3** |
| Perplexity.ai | - | 71.2 | - | - |
| *Baselines without retrieval* | | | | |
| LLaMA2$_{7B}$ | 14.7 | 44.5 | 34.2 | 21.8 |
| Alpaca$_{7B}$ | 23.6 | 45.8 | 49.8 | 45.0 |
| LLaMA2$_{13B}$ | 14.7 | 53.4 | 29.4 | 29.4 |
| Alpaca$_{13B}$ | 24.4 | 50.2 | 55.5 | 54.9 |
| CoVE$_{65B}$ | - | 71.2 | - | - |
| *Baselines with retrieval* | | | | |
| LLaMA2$_{7B}$ | 38.2 | 78.0 | 30.0 | 48.0 |
| Alpaca$_{7B}$ | 46.7 | 76.6 | 40.2 | 48.0 |
| SAIL | - | - | 69.2 | 48.4 |
| LLaMA2$_{13B}$ | 45.7 | 77.5 | 30.2 | 26.0 |
| Alpaca$_{13B}$ | 46.1 | 77.7 | 51.1 | 57.6 |
| *LLaMA2-hf-7b* | | | | |
| RAG | 50.5 | 44.9 | 48.9 | 43.4 |
| CRAG | **54.9** | 47.7 | **59.5** | **53.7** |
| Self-RAG* | 29.0 | 32.2 | 0.7 | 23.9 |
| Self-CRAG | 49.0 | **69.1** | 0.6 | 27.9 |
| *SelfRAG-LLaMA2-7b* | | | | |
| RAG | 52.8 | 59.2 | 39.0 | 53.2 |
| CRAG | 59.8 | 74.1 | **75.6** | 68.6 |
| Self-RAG | 54.9 | 81.2 | 72.4 | 67.3 |
| Self-CRAG | **61.8** | **86.2** | 74.8 | 67.2 |

## 5.2 BASELINES

We primarily compared CRAG with both approaches with and without retrieval, where the latter can be further split into standard RAG and latest advanced RAG, including:

**Baselines without retrieval.** We evaluated some public LLMs, LLaMA2-7B,13B (Touvron et al., 2023b), instruction-tuned models, Alpaca-7B,13B (Dubois et al., 2023), and CoVE$_{65B}$ (Dhuliawala et al., 2024) which introduces iterative engineering to improve the factuality of LLM generations. Propriety LLMs such as LLaMA2-chat$_{13B}$ and ChatGPT are also included.

**Standard RAG.** We evaluated the standard RAG (Lewis et al., 2020) where an LM generates output given the query prepended with the top retrieved documents using the same retriever as in our system. Here we adopted several public instruction-tuned LLMs, including LLaMA2-7B, 13B (Touvron et al., 2023b), Alpaca-7B,13B (Dubois et al., 2023), as well as LLaMA2-7B instruction-tuned in Self-RAG (Asai et al., 2024).

**Advanced RAG.** (1) SAIL (Luo et al., 2023) that instruction-tuned an LM on the Alpaca instruction-tuning data with top retrieved documents inserted before instructions. (2) Self-RAG (Asai et al., 2024) that tuned the LLaMA2 on the instruction-tuning data comtaining several sets of reflection tokens which were labeled by GPT-4 (OpenAI, 2023). (3) Following Asai et al. (2024), we also cited the results of retrieval-augmented baselines trained with private data: Ret-ChatGPT and Ret-LLaMA-chat, which deploy the same augmentation technique above, as well as perplexity.ai, an InstructGPT-based production search system.

## 5.3 RESULTS

Table 1 presents the results on four datasets. The model coupling the proposed method with standard RAG is named CRAG and that coupling with Self-RAG is named Self-CRAG. Readers can refer to Appendix B.3 for more implementation details of our proposed methods. From these results, we can conclude the following findings:

*First, the proposed method can significantly improve the performance of RAG and Self-RAG.* Specifically, as shown in table 1, CRAG outperformed RAG by margins of 19.0% accuracy on PopQA, 14.9% FactScore on Biography, 36.6% accuracy on PubHealth, and 8.1% accuracy on Arc-Challenge when based on *SelfRAG-LLaMA2-7b*, as well as by margins of 9.6% accuracy on PopQA, 2.8% FactScore on Biography, and 2.0% on Arc-Challenge when based on *LLaMA2-hf-7b*. Compared with the current state-of-the-art Self-RAG, Self-CRAG outperformed it by margins of 20.0% accuracy on PopQA, 36.9% FactScore on Biography, and 4.0% accuracy on Arc-Challenge when based on *LLaMA2-hf-7b*, as well as by margins of 6.9% accuracy on PopQA, 5.0% FactScore on Biography, and 2.4% accuracy on PubHealth, when based on *SelfRAG-LLaMA2-7b*. These results demonstrated the adaptability of CRAG which is plug-and-play and can be implemented into RAG-based approaches.

*Second, the proposed method demonstrated great generalizability across a variety of generation tasks.* In particular, these benchmarks reported in Table 1 respectively represent different practical scenarios including short-form entity generation (PopQA), long-form generation (Biography), and closed-set tasks (PubHealth, Arc-Challenge). These results verified the consistent effectiveness of CRAG. Its versatility across a spectrum of tasks underscores its robust capabilities and generalizability across diverse scenarios.

*Third, the proposed method exhibited greater flexibility in replacing the underlying LLM generator.* It can be seen that CRAG still showed competitive performance when the underlying LLMs was changed from *SelfRAG-LLaMA2-7b* to *LLaMA2-hf-7b*, while the performance of Self-RAG dropped significantly, even underperforming the standard RAG on several benchmarks. The reason for these results is that Self-RAG needs to be instruction-tuned using human or LLM annotated data to learn to output special critic tokens as needed, while this ability is not learned in common LLMs. CRAG does not have any requirements for this ability. As you can imagine, when more advanced LLMs are available in the future, they can be coupled with CRAG easily, while additional instruction tuning is still necessary for Self-RAG.

## 5.4 ABLATION STUDY

**The impact of each triggered action.** To further verify the effectiveness of triggered actions designed in the retrieval evaluator, ablation tests for removing each single action in the proposed method were conducted as shown in Table 2. Evaluations on the PopQA dataset were conducted to demonstrate the performance change in terms of accuracy. Specifically, when the action `Correct` or `Incorrect` was removed, it was merged with `Ambiguous` so that the proportion that originally triggered `Correct` or `Incorrect` would trigger `Ambiguous`.

Table 2: Ablation study for removing each single action on the PopQA dataset in terms of accuracy.

|  | LLaMA2-hf-7b | SelfRAG-LLaMA2-7b |
|---|---|---|
| CRAG | 54.9 | 59.8 |
| w/o. `Correct` | 53.2 | 58.3 |
| w/o. `Incorrect` | 54.4 | 59.5 |
| w/o. `Ambiguous` | 54.0 | 59.0 |
| Self-CRAG | 49.0 | 61.8 |
| w/o. `Correct` | 43.6 | 59.6 |
| w/o. `Incorrect` | 47.7 | 60.8 |
| w/o. `Ambiguous` | 48.1 | 61.5 |

On the other hand, when the action `Ambiguous` was removed, there was only one threshold against which all input queries clearly triggered `Correct` or `Incorrect`. From these results, it can be seen that there was a performance drop no matter which action was removed, illustrating that each action contributed to improving the robustness of generation. To further illustrate the study, experiments are also conducted by triggering only one action once, and the results shown in the appendix also prove the consistency.

**The impact of each knowledge utilization operation.** Table 3 illustrated how the performance changed if a key knowledge utilization operation was ablated. Evaluations on the PopQA dataset in terms of accuracy were conducted by individually removing the knowledge utilization operations of document refinement, search query rewriting, and external knowledge selection. Removing document refinement denoted that the original retrieved documents were directly fed to the following generator, as in most existing works. Additionally, removing

Table 3: Ablation study for removing each knowledge utilization operation on the PopQA in terms of accuracy.

|  | LLaMA2-hf-7b | SelfRAG-LLaMA2-7b |
|---|---|---|
| CRAG | 54.9 | 59.8 |
| w/o. refinement | 49.8 | 54.2 |
| w/o. rewriting | 51.7 | 56.2 |
| w/o. selection | 50.9 | 58.6 |
| Self-CRAG | 49.0 | 61.8 |
| w/o. refinement | 35.9 | 52.2 |
| w/o. rewriting | 37.2 | 58.4 |
| w/o. selection | 24.9 | 57.9 |

search query rewriting denoted that questions were not rewritten into queries consisting of keywords during knowledge searching. Eventually, removing knowledge selection denoted that all searched content of web pages was all regarded as the external knowledge without selection. These results help derive the findings that the performance of the final system degraded no matter which knowledge utilization operation was removed, revealing that each knowledge utilization operation contributed to improving the utilization of knowledge.

## 5.5 ACCURACY OF THE RETRIEVAL EVALUATOR

The quality of the retrieval evaluator significantly determined the performance of the entire system. Given the document retrieval results, we assessed whether the retrieval evaluator can accurately determine the overall quality of these results. The assessment accuracy on the PopQA dataset of our retrieval evaluator and the commercial LLM ChatGPT on the document retrieval results was shown in

Table 4: Evaluation of our retrieval evaluator and ChatGPT for the retrieval results on the PopQA dataset.

|  | Accuracy |
|---|---|
| Our Retrieval Evaluator (T5-based) | 84.3 |
| ChatGPT | 58.0 |
| ChatGPT-CoT | 62.4 |
| ChatGPT-few-shot | 64.7 |

Table 4. The prompts of *ChatGPT*, *ChatGPT-CoT*, and *ChatGPT-few-shot* used in our experiments can be referred to in Appendix A. Results reveal that the lightweight T5-based retrieval evaluator significantly outperformed the competitive ChatGPT in all settings.

## 5.6 ROBUSTNESS TO RETRIEVAL PERFORMANCE

To further verify the robustness of the proposed method to retrieval performance, we studied how the generation performance changed given different retrieval performance. A part of accurate retrieval results were deliberately removed at random to imitate a low-quality retriever and evaluate how the performance changed. Figure 3 demonstrated the performance change of Self-RAG and Self-CRAG on the PopQA dataset. It can be seen that the generation performance of Self-RAG and Self-CRAG dropped as the retrieval performance dropped, indicating that the generator relied heavily on the quality of the retriever. Furthermore, as the retrieval performance dropped, the

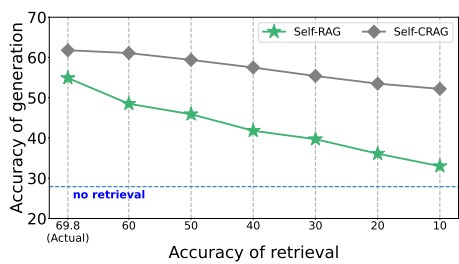

Figure 3: The generation performance of Self-RAG and Self-CRAG given different retrieval performance on the PopQA dataset with SelfRAG-LLaMA-7b. The lower horizontal line demonstrates the performance of the generator without retrieval.

generation performance of Self-CRAG dropped more slightly than that of Self-RAG. These results imply the superiority of Self-CRAG over Self-RAG on enhancing the robustness to retrieval performance.

### 5.7 CONSISTENT SUPPLEMENTATION OF WEB SEARCH KNOWLEDGE

This paper highlights the necessity of enhancing the retrieved context by incorporating additional information when the initial retrieval results are irrelevant and unreliable. Meanwhile, it is also crucial to confirm that the primary improvements in our method stem from the self-correction mechanism, rather than solely from the supplementary information obtained through web searches. To further demonstrate the effectiveness of the proposed self-correction mechanism, both RAG and Self-RAG were consistently supplemented

Table 5: Comparison results between CRAG, Self-CRAG and RAG, Self-RAG with the same input in terms of accuracy.

|  | LLaMA2-hf-7b | SelfRAG-LLaMA2-7b |
|---|---|---|
| PopQA |  |  |
| CRAG | 54.9 | 59.8 |
| RAG | 50.5 | 52.8 |
| RAG w. web | 52.2 | 53.8 |
| Self-CRAG | 49.0 | 61.8 |
| Self-RAG | 29.0 | 54.9 |
| Self-RAG w. web | 24.9 | 57.9 |

with web search knowledge to ensure they had access to the same scope of the retrieved knowledge. The results in Table 5 show that consistently supplementing RAG or Self-RAG with web search knowledge can improve the performance in most cases (except Self-RAG w. web using the original LLaMA2 model), though the improvement remains limited. Furthermore, augmenting RAG or Self-RAG with the proposed self-correction mechanism significantly outperformed the models consistently supplemented with web search knowledge in all cases. This finding confirms that the observed advancements are primarily attributable to the proposed self-correction mechanism.

### 5.8 COMPUTATIONAL OVERHEAD ANALYSIS

To illustrate that our self-correction mechanism serves as a lightweight, plug-and-play solution for various RAG-based frameworks, we measured the computational overhead. FLOPs prediction formulas in Narayanan et al. (2021) were employed, with the results presented in Table 6 which shows the predicted FLOPs per token on GPUs. Due to the adaptive nature of Self-RAG, which varies its generation strategies based on input, the computational overhead cannot be precisely determined. Therefore, we present an estimated range instead. Additionally, we conducted the experiments on PopQA to assess the average execution time per

Table 6: computational overhead assessment of RAG, CRAG, Self-CRAG, and Self-RAG about FLOPs per token on GPUs and executing time per instance. The upper bound of Self-CRAG is lower because only three passages are provided as input (correct, incorrect and ambiguous content). All the data in the table only represents a rough estimate of the generation phase, the retrieval and data-processing stages are not included.

|  | TFLOPs per token | executing time(s) |
|---|---|---|
| RAG | 26.5 | 0.363 |
| CRAG | 27.2 | 0.512 |
| Self-RAG | 26.5~132.4 | 0.741 |
| Self-CRAG | 27.2~80.2 | 0.908 |

instance in practice, as detailed in Table 6. The findings indicate that the self-correction mechanism incurs only modest computational overhead while significantly enhancing performance, thereby validating its lightweight nature.

## 6 CONCLUSION & LIMITATION

This paper studies the problem where RAG-based approaches are challenged if retrieval goes wrong, thereby exposing inaccurate and misleading knowledge to generative LMs. Corrective Retrieval Augmented Generation is proposed to improve the robustness of generation. Essentially, a lightweight retrieval evaluator is to estimate and trigger three knowledge retrieval actions discriminately. With the further leverage of web search and optimized knowledge utilization, CRAG has significantly improved the ability of automatic self-correction and efficient utilization of retrieved documents. Experiments extensively demonstrate its adaptability to RAG-based approaches as well as generalizability across short- and long-form generation tasks. While we primarily proposed to improve the RAG framework from a corrective perspective and CRAG can be seamlessly coupled with various RAG-based approaches, fine-tuning an external retrieval evaluator is inevitable. How to eliminate this external evaluator and equip LLMs with better retrieval evaluation capabilities will be our future work.

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

## A   TASK PROMPTS

The prompts for generating knowledge keywords as web search queries were illustrated in Table 7.

Table 7: The few-shot prompt to GPT-3.5 Turbo for generating knowledge keywords as web search queries.

Extract at most three keywords separated by comma from the following dialogues and questions as queries for the web search, including topic background within dialogues and main intent within questions.

question: What is Henry Feilden's occupation?
query: Henry Feilden, occupation

question: In what city was Billy Carlson born?
query: city, Billy Carlson, born

question: What is the religion of John Gwynn?
query: religion of John Gwynn

question: What sport does Kiribati men's national basketball team play?
query: sport, Kiribati men's national basketball team play

question: [question]
query:

The prompts to instruct ChatGPT as the evaluator were illustrated in Table 8, Table 9, and Table 10 respectively.

Table 8: The direct prompt to GPT-3.5 Turbo as the evaluator.

Given a question, does the following document have exact information to answer the question? Answer yes or no only.
Question: [question]
Document: [document]

Table 9: The prompt to GPT-3.5 Turbo with Chain-of-Thought as the evaluator.

Given a question, does the following document have exact information to answer the question?
Question: [question]
Document: [document]
Think Step by step, and answer with yes or no only.

Table 10: The few-shot prompt to GPT-3.5 Turbo as the evaluator.

Given a question, does the following document have exact information to answer the question? Answer yes or no only.

Question: In what city was Abraham Raimbach born?
Document: Bancroft was born on November 25, 1839 in New Ipswich, New Hampshire to James Bancroft and Sarah Kimball. At an early age he was cared for by Mr. and Mrs. Patch of Ashby, Massachusetts, the neighboring town. While not legally adopted, they named him Cecil Franklin Patch Bancroft, adding Franklin Patch after the son Mr. and Mrs. Patch had who recently died. He attended public schools in Ashby as well as the Appleton Academy in New Ipswich. He entered Dartmouth College in 1856 at the age of sixteen and graduated in 1860 near the top of his class. Bancroft continued his education as he began his career in teaching. He took classes at the Union Theological Seminary in New York City during the 1864-65 academic year. While there he was a member of the United States Christian Commission, traveling to support soldiers during the Civil War. He then transferred to the Andover Theological Seminary where he would graduate in 1867.
Answer: No.

Question: In what country is Wilcza Jama, Sokółka County?
Document: Wilcza Jama is a village in the administrative district of Gmina Sokółka, within Sokółka County, Podlaskie Voivodeship, in north-eastern Poland, close to the border with Belarus.
Answer: Yes.

Question: What sport does 2004 Legg Mason Tennis Classic play?
Document: The 2004 Legg Mason Tenis Classic was the 36th edition of this tennis tournament and was played on outdoor hard courts. The tournament was part of the International Series of the 2004 ATP Tour. It was held at the William H.G. FitzGerald Tennis Center in Washington, D.C. from August 16 through August 22, 2004.
Answer: Yes.

Question: Who is the author of Skin?
Document: The Skin We're In: A Year of Black Resistance and Power is a book by Desmond Cole published by Doubleday Canada in 2020. The Skin We're In describes the struggle against racism in Canada during the year 2017, chronicling Cole's role as an anti-racist activist and the impact of systemic racism in Canadian society. Among the events it discusses are the aftermath of the assault of Dafonte Miller in late 2016 and Canada 150. The work argues that Canada is not immune to the anti-Black racism that characterizes American society. Due to an error by the publisher, the initial printing of the book's cover did not include word Black in the subtitle. The mistake was later corrected. The book won the Toronto Book Award for 2020. In 2021, the book was nominated for the Shaughnessy Cohen Prize for Political Writing.
Answer: No.

Question: [question]
Document: [document]
Answer:

# B  Experiments

## B.1  Tasks, Datasets and Metrics

CRAG was evaluated on four datasets, which are in public domain and licensed for research purposes, including:

**PopQA** (Mallen et al., 2023) is a *short*-form generation task. Generally, only one entity of factual knowledge is expected to be answered for each single question. In our experiments, we exactly followed the setting in Self-RAG (Asai et al., 2024) which evaluated methods on a long-tail subset consisting of 1,399 rare entity queries whose monthly Wikipedia page views are less than 100. Accuracy was adopted as the evaluation metric.

**Biography** (Min et al., 2023) is a *long*-form generation task that is tasked to generate a detailed biography about a certain entity. Following previous work, FactScore (Min et al., 2023) was adopted to evaluate the generated biographies.

**PubHealth** (Zhang et al., 2023a) is a task in health care domain consisting of true-or-false questions. Claims are represented about health with factual information, and the model is tasked to verify the authenticity and give the judgment. Accuracy was adopted as the evaluation metric.

**Arc-Challenge** (Bhakthavatsalam et al., 2021) is a multiple-choice question task about some daily commonsense science phenomena. Given a scientific event that occurs in daily life, the model is required to select the correct description among 3 or 4 optional choices. Accuracy was adopted as the evaluation metric as well.

## B.2  Experiments compute Resources

We used NVIDIA A800 80GB GPU for experiments. For LLaMA-2 (7B) generation, it occupies over 40GB memory during inference. For T5-large (0.77B) fine-tuning, it takes much less compared with LLaMA-2.

## B.3  Implementation Details

**Retrieval Evaluator:** We fine-tuned the retrieval evaluator based on the lightweight T5-large (Raffel et al., 2020) pre-trained model. The dataset we used is the version provided by Self-RAG (Asai et al., 2024). Specifically, the original PopQA dataset consists of 14k samples, 1,399 of which were used for testing following Self-RAG (Asai et al., 2024), and the remaining were used for fine-tuning to avoid information leakage. Besides, the fine-tuned evaluator was transferred and also utilized on the Bio, Pub and ARC datasets during inference. The label of positive samples was 1, while that of negative ones was -1. At inference, the evaluator scored the relevance from -1 to 1 for each document. The two confidence thresholds for triggering one of the three actions were set empirically. Specifically, they were set as (0.59, -0.99) in PopQA, (0.5, -0.91) in PubQA and Arc-Challenge, as well as (0.95, -0.91) in Biography.

**Internal Knowledge:** To obtain fine-grained retrieval results, we segmented the retrieved results into internal strips. If a retrieved result is as short as one or two sentences, it is regarded as an individual strip, otherwise, retrieval documents are required to be split into smaller units which generally consist of a few sentences according to the total length. The scale is assumed to include an independent piece of information, and the filtering is based on the segments. We directly adopted the evaluator again for knowledge strips filtering, and the top-k is set to 5, filter threshold as -0.5.

**External Knowledge:** Google Search API was adopted to search for the relevant URLs, top-k is set to 5, and pages from Wikipedia will be added preferentially. The searched web pages are generally in the form of HTML files, where content is split with special tokens like <p> and </p>. Thus an extra segmentation like the knowledge refinement is not required, related knowledge paragraphs can be directly selected with the evaluator similar to internal knowledge. In this way, the accuracy of the search outcomes can be ensured without compromising the quality and relevance of the information used for generation.

**Generator:** As CRAG is a plug-and-play method, all generation models that can be utilized in RAG fit our approach as well. To be consistent with baselines for comparison, we adopted LLaMA2

(Touvron et al., 2023b) for the generation. We first introduced the *LLaMA2-hf-7b* from huggingface to generate responses. Since Self-RAG (Asai et al., 2024) fine-tuned LLaMA2 and reached a new state-of-the-art performance on several tasks, we further utilized the launched model, *SelfRAG-LLaMA2-7b*, as a new generator to be consistent with their work and study the specific improvement of our method.

**Self-CRAG:** To demonstrate that our plug-and-play approach can be utilized in other concurrent studies, we specifically designed to insert our CRAG into the Self-RAG (Asai et al., 2024) framework and named it Self-CRAG. Self-RAG is an advanced RAG approach that introduces a critic model to decide whether to retrieve and which retrieved document to be referred for generation. It meets our demand for deciding which action to be triggered, thus we replaced the retrieved items in Self-RAG with our processed internal knowledge for `Correct`, external knowledge for `Incorrect`, and combined knowledge for `Ambiguous`.

### B.4    MORE DETAILED RESULTS

**Ablation Study:** The following results in Table 11 demonstrate the ablation study by triggering one action only for all instances.

### B.5    RESULTS ON PUBHEALTH AND ARC-CHALLENGE

It is worth mentioning that the performance on PubHealth based on *LLaMA2-hf-7b* was much worse than others. We studied these cases and found that *LLaMA2-hf-7b* is relatively weak in instruction comprehension. Most of the cases fail to generate `True` or `False` in such a binary-question task, resulting in a low accuracy during the evaluation. This situation somewhat happens in Arc-Challenge as well, when the model is tasked to generate the index of a candidate.

Table 11: Ablation study for removing only a single action on the PopQA dataset in terms of accuracy.

|  | LLaMA2-hf-7b | SelfRAG-LLaMA2-7b |
|---|---|---|
| CRAG | 54.9 | 59.8 |
| only `Correct` | 52.4 | 56.7 |
| only `Incorrect` | 47.0 | 48.5 |
| only `Ambiguous` | 52.7 | 58.0 |
| Self-CRAG | 49.0 | 61.8 |
| only `Correct` | 48.6 | 57.2 |
| only `Incorrect` | 40.8 | 53.3 |
| only `Ambiguous` | 44.9 | 59.8 |

