# OpenReview forum: "Corrective Retrieval Augmented Generation"
_ICLR.cc/2025/Conference — ICLR 2025 Conference Withdrawn Submission_

### Official Review · Reviewer_Yars · 2024-10-27

**Soundness:** 2
**Presentation:** 2
**Contribution:** 1
**Rating:** 3
**Confidence:** 4

**Summary:**

The paper presents Corrective Retrieval Augmented Generation (CRAG), a method aimed at improving the robustness of retrieval-augmented generation (RAG) in large language models (LLMs). CRAG introduces a lightweight retrieval evaluator to assess the quality of retrieved documents and determine appropriate actions: correct, incorrect, or ambiguous. The method leverages large-scale web searches and a decompose-then-recompose algorithm to refine retrieved information.

**Strengths:**

1. The presentation is clear and easy to follow.

2. The paper presents comprehensive experiments across multiple datasets to show the performance of the proposed method.

**Weaknesses:**

1. The technical contribution of this paper is limited: The proposed method brings limited technical contribution to RAG area. Components of the proposed method are similar to by existing methods in Information Retrieval or RAG. For example, assessing the quality of retrieved texts to determine whether retrieval or not has been fully studied in existing RAG methods such as Self-RAG [1] and RetRobust [2]. Using large-scale web resources to replace the static corpus cannot be seen as an technical contribution of this paper. I cannot learn any insights that are distinguishable from many existing RAG papers from this paper.

2. Unfair comparison: Why CRAG uses large-scale web resources but many baselines not, such as Self-RAG.

**Questions:**

What insights can we learn from this paper? Many components of the proposed method are similar to existing methods in RAG and Information Retrieval.

---

### Official Review · Reviewer_aMN7 · 2024-10-29

**Soundness:** 3
**Presentation:** 3
**Contribution:** 2
**Rating:** 6
**Confidence:** 4

**Summary:**

This paper proposes the Corrective Retrieval Augmented Generation (CRAG) framework to enhance the robustness of large language model (LLM) generation. The model aims to improve the quality of retrieved documents through a lightweight retrieval evaluator and large-scale network searches. CRAG enhances text robustness by self-correcting retrieval results and has been validated across multiple datasets through extensive experiments.

**Strengths:**

1. Compared to Standard RAG, CRAG shows significant improvements, although experiments were only conducted on a 7B model.
2. Knowledge Correction is necessary as it can assess the accuracy of retrieval results and prevent irrelevant results from impacting model performance.
3. The paper includes numerous ablation studies, making the experiments overall solid.

**Weaknesses:**

1. A core concern is latency; the introduction of the Knowledge Correction phase significantly increases delays, which CRAG does not discuss. The authors only discuss the time consumption of the generation phase, which could hinder CRAG's practical application.
2. Additionally, the accuracy of the T5-based Retrieval Evaluator is concerning. It is also unclear how the evaluator's accuracy impacts CRAG's generation.
3. The lack of case analysis is notable. It would be beneficial to understand how different labels {Correct, Incorrect, Ambiguous} affect CRAG's generation outcomes and how changes in these labels alter the results.

**Questions:**

1. If the model is sufficiently strong, could it internally differentiate between accurate and inaccurate retrieval results without needing a Knowledge Correction process? I suspect that adding experiments with (CRAG + ChatGPT) and (CRAG + LLaMA 70B) might be necessary

---

### Official Review · Reviewer_xAjx · 2024-11-01

**Soundness:** 1
**Presentation:** 2
**Contribution:** 2
**Rating:** 3
**Confidence:** 5

**Summary:**

The paper introduces Corrective Retrieval Augmented Generation (CRAG) to reduce hallucinations in large language models (LLMs). CRAG improves generation robustness by assessing retrieved document quality with a confidence score, triggering corrective actions if needed. It also enhances retrieval by adding large-scale web searches and a decompose-then-recompose algorithm to filter key information from documents. This plug-and-play method boosts RAG-based performance, as shown by experiments across multiple datasets.

**Strengths:**

1. The paper addresses an important research question that is critical. Recognizing the imperfections in existing Retriever technologies, this paper focuses on how to mitigate these issues within the Retrieval-Augmented Generation (RAG) paradigm, thus contributing valuable insights toward developing more robust RAG systems.

2. The paper presents compelling experimental results.

**Weaknesses:**

1. **The experiment in this paper is not sufficient, particularly in the selection of baselines.**

The paper lacks comparison with highly relevant prior works that have proposed corrective strategies for Retrieval-Augmented Generation (RAG). Notably, it does not include baselines such as RARR [1] and DRAGIN [2], which are essential for contextualizing the contribution of this work within the existing literature. Including these baselines would strengthen the evaluation and provide a clearer understanding of the advancements offered by the proposed method.

Furthermore, the baselines used for comparison are inadequate, with a noticeable absence of state-of-the-art (SOTA) baselines from 2024. For a comprehensive and fair evaluation, it is imperative to compare the proposed method against at least two or three contemporary baselines from 2024. This would demonstrate the method’s effectiveness relative to the latest advancements in the field.

2. The ablation results indicate that removing each component of the proposed method has minimal impact on the final results. This suggests that the individual components may not significantly contribute to the overall performance. A deeper analysis is needed to understand the role and necessity of each component within the model.

3. The paper states:

>“To the best of our knowledge, this work makes the first attempt to design corrective strategies for RAG to improve its robustness.”

This claim is inaccurate. Prior works such as RARR [1] and Dragin [2] have already explored corrective strategies for RAG. It is important for the authors to accurately acknowledge existing research to position their contribution correctly within the academic discourse.


4. (Minor) The paper appears to have been prepared in a hurry, there are formatting errors such as the question mark in the citation on Line 98.

5. About methodology

The overall methodology and structure of the paper seem disorganized. The motivation presented focuses on addressing the reliance on the relevance of retrieved documents and correcting instances where the retriever returns incorrect information. However, several components of the proposed solution do not directly address this motivation. For instance, incorporating Web Search when retrieval results are poor raises questions. If a superior retriever (Web Search) is available within the pipeline, it is unclear why it is not utilized from the beginning. Additionally, the inclusion of passage splitting does not seem directly related to correcting retrieval errors, leading to confusion about how these components align with the stated goals. The training methodology and performance metrics of the retrieval evaluator are not adequately described. It is unclear how the evaluator is trained and assessed. The reliance on existing datasets like PopQA for relevance signals may introduce biases, as these datasets might not accurately reflect real-world retrieval tasks. Moreover, utilizing T5-Large, which is an older and less capable model compared to current large language models, may limit the evaluator’s effectiveness, particularly for complex queries.

6. There is a lack of transparency regarding whether the baselines use the Web Search API. If the proposed method leverages advanced commercial search engines while the baselines rely solely on a single retrieval method, the comparison is inherently unfair.


References

[1] RARR: Researching and Revising What Language Models Say, Using Language Models

[2] Dragin: Dynamic Retrieval Augmented Generation Based on the Real-time Information Needs of Large Language Models

**Questions:**

Please refer to the weakness section

---

### Official Review · Reviewer_4nYX · 2024-11-01

**Soundness:** 2
**Presentation:** 3
**Contribution:** 2
**Rating:** 3
**Confidence:** 5

**Summary:**

This paper proposes a CRAG framework to enhance the quality of retrieved documents in Retrieval-Augmented Generation (RAG). Specifically, a small LLM is employed to evaluate the relevance of retrieved documents and take different actions based on its confidence scores. Additionally, document refinement and web search are incorporated to improve the quality of the retrieved documents.

**Strengths:**

1. Authors propose a cost-effective method to filter, refine and improve documents in RAG. This method can be applied in various RAG pipelines.

2. Detailed experiments show the effectiveness, robustness, efficiency of CRAG.

**Weaknesses:**

1. CRAG can not handle the multi-hop QA task. Since the confidence score is calculated between a query and a single document, the evaluator is hard to judge the complex relationships within multiple documents.
2. The filtering and refinement processes depend on the evaluator, which can potentially filter out the useful information. Besides, the paper lacks an evaluation and analysis of the filtering performance. What if the evaluator filters out helpful results?
3. It is challenging to manually determine the threshold in real-world scenarios. As the thresholds settings shown in Appendix B.3, it can vary significantly across different datasets.

**Questions:**

1. The utilization of the small LLM (T5 in authors’ implementation) is weird. It seems that an encoder-based classifier can do the evaluator’s job. Also, we can obtain the scores from a retriever (even not fine-tuned) by calculating similarity between query and document embeddings.
2. It is unclear to include the results of LMs trained on proprietary data in Table 1. It appears that the comparison between the authors' method and baseline methods is only reflected on the LLaMA2 and Self-RAG LLaMA2.
3. In Section 5.3, the accuracy improvements of CRAG over RAG do not match the data in Table 1. Are these typos or wrong results?
4. All results in the paper are all reported on LLaMA2 7B models. As CRAG is a flexible method, it would be beneficial for authors to conduct experiments on stronger LLMs (such as LLaMA3 or ChatGPT) to augment its suitability and effectiveness.
5. What is the proportion of different actions during inference?

---

### Note · Authors · 2024-11-26

I have read and agree with the venue's withdrawal policy on behalf of myself and my co-authors.